# Assessing the Efficacy of a Virtual Assistant in the Remote Cardiac Rehabilitation of Heart Failure and Ischemic Heart Disease Patients: Case-Control Study of Romanian Adult Patients

**DOI:** 10.3390/ijerph20053937

**Published:** 2023-02-22

**Authors:** Andreea-Elena Lăcraru, Ștefan-Sebastian Busnatu, Maria-Alexandra Pană, Gabriel Olteanu, Liviu Șerbănoiu, Kai Gand, Hannes Schlieter, Sofoklis Kyriazakos, Octavian Ceban, Cătălina Liliana Andrei, Crina-Julieta Sinescu

**Affiliations:** 1Department of Cardiology, University of Medicine and Pharmacy “Carol Davila”, Emergency Hospital “Bagdasar-Arseni”, 050474 Bucharest, Romania; 2Research Group Digital Health, Faculty of Business and Economics, Technische Universitat Dresden, 01062 Dresden, Germany; 3Department of Business Development and Technology, Aarhus University, 7400 Aarhus, Denmark; 4Economic Cybernetics and Informatics Department, University of Economic Studies, 010374 Bucharest, Romania

**Keywords:** telerehabilitation, virtual assistants, cardiac rehabilitation

## Abstract

Cardiovascular diseases (CVDs) are the leading cause of mortality in Europe, with potentially more than 60 million deaths per year, with an age-standardized rate of morbidity-mortality higher in men than women, exceeding deaths from cancer. Heart attacks and strokes account for more than four out of every five CVD fatalities globally. After a patient overcomes an acute cardiovascular event, they are referred for rehabilitation to help them to restore most of their normal cardiac functions. One effective way to provide this activity regimen is via virtual models or telerehabilitation, where the patient can avail themselves of the rehabilitation services from the comfort of their homes at designated timings. Under the funding of the European Union’s Horizon 2020 Research and Innovation program, grant no 769807, a virtual rehabilitation assistant has been designed for elderly patients (vCare), with the overall objective of supporting recovery and an active life at home, enhancing patients’ quality of life, lowering disease-specific risk factors, and ensuring better adherence to a home rehabilitation program. In the vCare project, the Carol Davila University of Bucharest (UMFCD) was in charge of the heart failure (HF) and ischemic heart disease (IHD) groups of patients. By creating a digital environment at patients’ homes, the vCare system’s effectiveness, use, and feasibility was evaluated. A total of 30 heart failure patients and 20 ischemic heart disease patients were included in the study. Despite the COVID-19 restrictions and a few technical difficulties, HF and IHD patients who performed cardiac rehabilitation using the vCare system had similar results compared to the ambulatory group, and better results compared to the control group.

## 1. Introduction

### 1.1. Epidemiology of Cardiovascular Disease and the Current Status of Cardiac Rehabilitation

Coronary heart disease, cerebrovascular disease, rheumatic heart disease, and other illnesses are within the category of heart and blood vessel disorders known as cardiovascular diseases (CVDs). According to the World Health Organization (WHO), CVDs are the leading cause of mortality in Europe, with potentially more than 60 million deaths per year, with an age-standardized rate of morbidity-mortality higher in men than women, exceeding deaths from cancer. Heart attacks and strokes account globally for more than four out of every five CVD fatalities [1,2,3].

After a patient overcomes an acute cardiovascular event, they are referred for rehabilitation to help them restore most of their normal cardiac functions. That is where cardiovascular rehabilitation comes in. Cardiovascular rehabilitation (CR) is a customized program that combines medical therapy and physical activity with medical education, in order to accelerate recovery and enhance the health status in people with heart disease [4]. The 2021 European guidelines’ recommendations on CVD prevention in clinical practice underline that medication adherence and lifestyle modifications are crucial in secondary CVD prevention, which may be expanded and improved via CR programs, lowering the incidence of recurrent heart disease and the risk of overall mortality [5].

After the outbreak of the COVID-19 pandemic, there has been a great burden on hospitals and outpatient departments to cater to the needs of all patients appropriately and in a satisfactory manner, while respecting the conditions for preventing the new coronavirus from spreading. As a result, CR programs have been suspended, partly or altogether, in numerous hospitals and specialized rehabilitation clinics. Exercise testing and group-based exercise training programs have been difficult to implement during this period [6,7]. Despite all these difficulties that the pandemic has brought, the European Society of Cardiology and the American Heart Association maintain the recommendation of cardiac rehabilitation after myocardial infarction (MI), coronary artery bypass graft (CABG), and percutaneous coronary intervention (PCI), and find CR to be an important step towards the complete recovery of affected patients [8,9,10,11,12].

### 1.2. Virtual Assistants as an Alternative to Classic Recovery Programs

Despite the relaxation of COVID-19 restrictions, its presence still lingers in our society, and we must look after patients who need CR. One effective way to provide this activity regimen is via virtual models or telerehabilitation, where the patient can avail themselves of rehabilitation services from the comfort of their homes, at designated times. In 2021, in Belgium, 52% of rehabilitation facilities already offered CR programs with the use of telerehabilitation. Physicians, nurses, dieticians, physiotherapists, and psychologists were all engaged in remote CR application. Exercise instruction, nutrition counselling, smoking cessation assistance, cardiovascular education, psychological support, medication adherence, and weight management were among the CR components that were carried out [13,14,15].

It would be safe to say that virtual rehabilitation is not an unknown concept, and providing virtual care to affected patients from the comfort of their homes is possible, as noted in a stroke study conducted in 2018. These needs could be met by virtual reality (VR). As a more engaging and motivational tool, game-based treatments may increase patients’ involvement in rehabilitation therapy. Mobile devices may provide customized home-based treatment, with interactive contact between patients and therapists, such as cellphones, tablets, and personal computers, which can provide much-needed therapy and rehabilitation to the patients, within their comfort zone [16,17].

Despite the benefits of telerehabilitation, demonstrated in multiple studies, there are also data that cast doubt, which leaves a gap for us to explore whether virtual services, in the form of telerehabilitation or video games, provide the sufficient care that a patient needs after an acute cardiovascular or neurological event [18].

In 2017, under the funding of the European Union’s Horizon 2020 Research and Innovation program, grant no 769807, a virtual rehabilitation assistant has been designed for elderly patients, with the overall objective of supporting recovery and an active life at home, enhance patients’ quality of life, lowering disease-specific risk factors, and ensuring better adherence to a home rehabilitation program. The Virtual Coaching Activities for Rehabilitation in Elderly (vCare) project has included partners from seven European countries, with the necessary technical and medical specialization for its construction. The entire project lasted five years, ending in August 2022 [19].

The primary goal of the project was to build and implement the virtual assistant in patients’ homes and to remotely monitor the individualized rehabilitation program that each of them received. Medical recovery areas included stroke (SD), Parkinson’s disease (PD), heart failure (HF), and ischemic heart disease (IHD) [19].

The secondary aim of the project was to evaluate the effectiveness of the remote recovery program. Contextually, part of the project took place during the COVID-19 pandemic, which made it difficult on one hand to implement the project in the patients’ homes, but on the other hand highlighted the importance of such a system in the patients’ lives [19].

In the vCare project, the Carol Davila University of Bucharest (UMFCD) was in charge of the heart failure and ischemic heart disease use cases.

## 2. Materials and Methods

### 2.1. Study Design and Objectives

We conducted a prospective interventional pilot study within Carol Davila University of Medicine and Pharmacy, Bucharest, on patients diagnosed with chronic heart failure and ischemic heart disease.

The primary objective of the study was to evaluate the use, feasibility, and effectiveness of the vCare system in motivating heart failure and ischemic heart disease patients to actively engage in a personalized cardiac rehabilitation program, in order to improve independence and quality of life.

The secondary objectives evaluated were the impact of the virtual system on the reduction of cardiovascular risk factors, on the adherence of patients to the rehabilitation plan of care, and on the personalization of treatment and promotion of an active life at home. The user experience was also evaluated through standardized questionnaires, with the aim of improving the system in the future with the help of user perception.

The virtual coaching system aimed to support the patient based on his/her needs, both in the home rehabilitation process (continuity of care) and in everyday activities, in order to promote an active and healthy lifestyle and to reduce the worsening of the disease and/or the occurrence of dangerous events (e.g., decompensation of heart failure). Data from various devices were processed and transformed into useful information for the patient.

### 2.2. Profile of Enrolled Patients

The recruitment period of the subjects was between 6–12 weeks and took place in the cardiology clinic of UMFCD within Bagdasar–Arseni Emergency Clinical Hospital in Bucharest. At the end of this period we managed to enroll 30 patients with chronic heart failure and 20 patients with ischemic heart disease. The inclusion and exclusion criteria used for enrollment in the study are presented in Table 1.

Considering that all patients eligible for enrollment in the project would have benefited from a cardiac rehabilitation program, we considered that the most equitable method of dividing them into study subgroups was by randomization.

After enrollment, the heart failure study group was randomly divided into: the vCare experimental subgroup (EG), who performed cardiac rehabilitation at home using the vCare app; the ambulatory subgroup (AG), who underwent conventional cardiac rehabilitation at Bagdasar–Arseni Hospital, in Bucharest; and the control subgroup (CG), who only received advice at discharge on how to perform a cardiac rehabilitation program at home. Consequently, the ischemic heart disease study group was randomly divided into: the vCare experimental subgroup (EG), who performed cardiac rehabilitation at home using the vCare app; and the control subgroup (AG), who only received advice at discharge on how to perform a cardiac rehabilitation program at home. Each of the presented subgroups had an equivalent number of 10 patients. The study took place over a period of three months for both use cases: heart failure (HF) and ischemic heart disease (IHD). After this period, the analysis and interpretation of the collected data, and the outline of the obtained results, were carried out, an action that stretched over a period of two months.

The enrollment of a small number of patients in this project was essential for testing the implemented system. As expected, technical errors occurred during the study, requiring them to be physically addressed by a technical team, which would have been difficult to achieve on a large group of patients. The implementation of this project overlapped with the COVID-19 pandemic, which led to the emergence of various difficulties (patient refusal, delay in the delivery of equipment, fear on the part of patients). Because of these problems, the cardiac recovery period of some patients in the experimental groups was less than 3 months, which may have adversely influenced the results obtained.

### 2.3. Ethical Considerations

The study groups consisted of patients who met the inclusion criteria and did not present any exclusion criteria, and who, prior to enrollment, underwent a detailed presentation of the study and training and guidance in the use of the equipment provided by the research team. The protection of patient data was carried out by means of the pseudo-anonymization method as follows: each patient received an identifier, with the purpose of separating personal data from those collected in the study. Also, each study participant was informed about the confidentiality, the collecting method, and the use of personal data. The research study presented no potential physical, legal, or psychological risk and was conducted in accordance with the ethical principles of the Helsinki Declaration of Human Rights. Each enrolled patient signed an informed consent for voluntary participation.

### 2.4. Initial Assessment and Structuring of the Virtual Environment

The clinical and paraclinical examination protocol was the same for each use case. Each patient underwent a clinical examination at the time of enrollment (T_0_), the collection of blood samples (blood count, ionogram, creatinine level, liver enzyme levels lipid profile, blood sugar, uric acid, creatine kinase level, erythrocyte sedimentation rate, fibrinogen, urine test), echocardiography, a cardio-pulmonary exercise test (for the assessment of exercise capacity and personalization of the cardiac rehabilitation program), a 24 h blood pressure monitoring, a 24 h electrocardiogram monitoring (to rule out uncontrolled blood pressure and potentially fatal arrhythmias), and psychological evaluation by means of standardized questionnaires (Minnesota, HADS scale, Fagerstrom, EuroQol-5D, EQ-VAS). With the note that the Minnesota questionnaire has been applied only to heart failure patients (Table 2).

Each patient of both vCare experimental groups (HF and IHD) received a toolkit that structured the virtual assistant digital environment. The devices that led to the structuring of this digital environment are presented in Table 3. The role, and collected parameters, of each of the devices used are presented in Table 4.

### 2.5. 3-Month Assessment and Collected Parameters

At the end of the rehabilitation program (T_1_), all patients were re-evaluated following the same examination protocol from baseline. The subjects allocated in the experimental group were additionally examined with three other evaluation scales, aiming to rate the usability perceived with the vCare components during their experience in the pilot trial (user experience questionnaire, system usability scale, technology acceptance model) (Table 5).

Standardized psychological and usability questionnaires were used in the study due to the increased reliability, validity, sensitivity, and objectivity of the answers provided.

The evaluation of the enrolled patients led to the collection of demographic, social, clinical, biological, and imaging parameters, and to a quantification of the cardiovascular risk factors present in the study groups. Along with them, the answers to the standardized questionnaires applied were collected and indexed. Parameters recorded by the virtual assistant were automatically collected and indexed for each patient of the two experimental groups.

### 2.6. Virtual Assistant Components

The cardiac rehabilitation program offered by the virtual assistant consisted of the components that are presented in Table 6, alongside the services provided. Apart from the alcohol reduction component, which has been designed and used only for the IHD study group, these were common to both study groups (Table 6).

The vCare virtual assistant was controlled remotely by the medical research team through a platform called Kiola, with access restricted to medical staff only. When the patient was enrolled in the remote cardiac recovery program, their medical data were entered and the cardiac recovery program was established by the cardiac rehabilitation team of specialists. Once active, it was found on the patient’s tablet at home, and the virtual assistant would guide them step by step in order to carry out the recommended daily activities. Each component of the vCare program had the capacity of automatic self-regulation through positive or negative feedback received from the patient, its positive or negative evolution, and/or the vital parameters monitored. The Kiola platform also had the role of storing all the recorded parameters, in order that they could be visualized at any time by the medical team.

### 2.7. Statistical Analysis

The UMFCD team entered the collected parameters from the HF and IHD patients who were recruited in the study into a Microsoft Excel database. Input variables were of type integer, real, or Boolean and were analyzed and studied individually. In order to draw conclusions regarding the improvements, distributions, and means of values for each subgroup, before and after values were compared. The t-test was used to determine if there were any changes between the values recorded at T_0_ (prior to initiation of the cardiac rehabilitation program) and T_1_ (after the conclusion of the rehabilitation program), and if these differences were by chance or had statistical significance. There were not missing values in the dataset. Finally, Python 3.8.5 was used to aggregate the findings, refine the data, and create tables.

## 3. Results

### 3.1. Heart Failure Participants

During the recruitment period, 47 initially eligible patients were identified for enrollment in the study. After clinical and paraclinical evaluation, 13 patients were excluded, due to the fact that their medical status did not allow them to enroll in a cardiac rehabilitation program. Thirty-four patients remained, of which four did not have the possibility of internet connection and/or a TV with an HDMI port at home. They were also excluded, thus forming a group of 30 patients. The 30 patients were randomly divided into the three previously mentioned subgroups and, out of the 10 patients of the experimental subgroup, one patient dropped out of the study, stating that there was too little space at home to carry out the exercises. However, the patient who dropped out performed the final assessment of the study, so their data were taken into account in the final analysis.

A total of 30 heart failure patients (17/13 M/F; 61.53 ± 9.41 years) were included in the study. The gender distribution in the EG and AG was five male patients and five female patients each, and in the CG there were six male patients and four female patients. The area of origin of the enrolled patients was mostly urban, with only one rural patient in EG, two rural patients in AG, and four rural patients in CG. From an educational point of view, most of the enrolled subjects had a high educational level, and only seven had an inferior educational level. From the point of view of financial status, 17 patients had average incomes, seven high incomes, and six patients had low incomes, evaluated at the current financial status of Romania.

Cardiovascular risk factors were evaluated in the study group and the following were identified, out of 30 patients: 10 were smokers; 21 were overweight, three had grade I obesity, and six had normal weight; 18 were identified with a sedentary level of physical activity; 28 were hypertensive, 10 patients had type 2 diabetes miellitus, and 18 had dyslipidemia.

The evaluation of exercise capacity, measured by VO_2max_ at T_0_ and T_1_, shows a statistically significant difference between the pre- and post-intervention groups’ values. Thus, an improvement in exercise capacity can be observed in both EG and AG, from 19.21 mL/kg/min and 19.18 mL/kg/min, to 21.32 mL/kg/min and 21.98 mL/kg/min, respectively. On the other hand, in the CG the VO_2max_ decreased from 18.46 mL/kg/min to 17.04 mL/kg/min (Figure 1).

According to Figure 2, between T_0_ and T_1_, LDL-cholesterol levels decreased in both experimental and ambulatory groups. The LDL-cholesterol improvement was much more pronounced in the EG, the decrease being approximately 30% of the initial value. The CG did not show significant changes between T_0_ and T_1_ (Figure 2).

In terms of quality of life, in the vCare and ambulatory groups an improvement was observed, along with a decrease in anxiety and depression levels. Patients enrolled in the EG and AG showed significantly improved results in both the MLFQH questionnaire and on the HAD scale. Unlike them, in the CG no improvement in quality of life and anxiety levels was observed, but the severity of the depression level increased (*p*-value = 0.4) (Figure 3, Figure 4 and Figure 5).

All smoking patients in the heart failure study group were assessed by means of the Fagerstrom questionnaire at the beginning of the cardiac rehabilitation program. In the EG, we enrolled three smokers, out of which only one was considered a regular smoker. Through the intervention of the virtual assistant, the patients in the EG presented a 50% reduction in nicotine use. In the AG, the decline in tobacco use was similar in terms of impact. In the CG, all four patients who were identified as smokers continued to smoke, but they also showed reductions in the number of cigarettes consumed per day (Figure 6).

### 3.2. Ischemic Heart Disease Pilot Results

During the recruitment period, 34 eligible patients were identified for enrollment in the study. After clinical and paraclinical evaluations, six patients were excluded due to the fact that their medical status did not allow them to enroll in a cardiac rehabilitation program. Twenty-eight patients remained, of which eight did not have the possibility of internet connection at home, and/or a TV with an HDMI port at home, and/or found the virtual assistant too difficult to use, according to the initial presentation. They were excluded, thus forming a group of 20 patients. The 20 patients were randomly divided into the two previously mentioned subgroups and, out of the experimental subgroup, two patients dropped out of the study, due to technical errors that occurred during the cardiac recovery program. However, the patients that dropped out performed the final assessment of the study, so their data were considered in the final analysis.

A total of 20 ischemic heart disease patients (16/4 M/F; 58.1 ± 7.12 years) were included in the study. The gender distribution was even between the two subgroups, with eight male and two female patients in each. The area of origin of the enrolled patients was mostly urban, with two rural patients in the EG and one rural patient in the CG. From the educational point of view, most of the enrolled patients had a high educational level and only three patients had inferior educational level. From the point of view of financial status, 11 patients had average incomes, four high incomes, and five low incomes, evaluated at the current financial status of Romania.

Cardiovascular risk factors were evaluated in the study group and the following were identified, out of 20 patients: seven were smokers; 12 were overweight, five patients had grade I obesity, and three patients had normal weight; 17 patients were identified with a sedentary level of physical activity; 20 patients were hypertensive, seven patients had type 2 diabetes miellitus, and 17 patients had dyslipidemia.

The effort capacity of the IHD study group was evaluated equivalently by means of the VO_2max_ parameter of the cardiopulmonary test. The patients enrolled in the EG improved their VO_2max_ level from 16.58 mL/kg/min to 20.7 mL/kg/min, while the ones in the CG showed a decrease of 1.75 mL/kg/min from T_0_ to T_1_ (Figure 7).

Regarding the assessment of dyslipidemia, both EG and CG showed improvements of LDL-cholesterol levels. In this regard, the CG showed a greater decrease, from 165.34 mg/dL to 132.42 mg/dL, but without reaching the target values for an ischemic heart disease patient. In the EG group a decrease of only 21.05 mg/dL from baseline was observed. (Figure 8).

The anxiety and depression levels in the IHD study group were assessed using the HAD scale. The anxiety level did not undergo any change in the CG during the study period, while a slight increase was observed in the EG. On the other hand, a significant improvement was assessed in depression level, with a much greater decrease in EG compared to CG (Figure 9).

At the end of the cardiac rehabilitation program, in the EG only one out of four smokers continued to use tobacco, but with a reduction in the addiction level, quantified by the number of daily consumed cigarettes. In the CG, two patients quit smoking, and one continued, with the same level of addiction as before the study period (Figure 10).

### 3.3. Quality of Life Assessment

#### 3.3.1. Heart Failure Quality of Life Results

EuroQol-5D assesses five parameters considered as defining factors for a high quality of life (mobility, pain, anxiety, self-care, and daily activities). Between T_0_ and T_1_ a considerable improvement was evaluated in the EG and AG of the HF study groups, with superior results in the latter. In the CG, the quality of life did not undergo significant changes, with the parameters undergoing statistically insignificant changes (Figure 11).

In the self-assessment quality of life questionnaire, the results were slightly different, with a small increase in the EG results and a more important one in the AG. An important discrepancy to mention is in the CG results, which highlight different scores than those of the EuroQol-5D. This difference may underline the importance of using multiple analysis parameters in the health status of patients (Figure 12).

#### 3.3.2. Ischemic Heart Disease Quality of Life Results

The patients enrolled in the EG of the IHD study groups were assessed with an increase in quality of life at the end of the cardiac rehabilitation program via the vCare app, while the quality of life of the CG patients remained the same between T_0_ and T_1_ (Figure 13).

Regarding the application of the EQ-VAS questionnaire to the IHD study groups, the results were consistent with those of the EuroQol-5D. Consequently, patients from the EG showed a 15% increase in quality of life between T_0_ and T_1_, while in the CG there were no changes observed (Figure 14).

An insight into how the parameters recorded by the virtual assistant are stored is shown in Table 7 and Table 8. These are examples of monitoring the number of steps and active weeks parameters within the cardiac recovery program. In a similar manner, all parameters recorded by the virtual assistant were stored and found on the Kiola platform, where they were analyzed by the medical team. It is important to mention that each parameter was individualized for each patient, so that the adherence may be different depending on the preset target for each one (Table 7 and Table 8).

### 3.4. Usefulness of vCare system

In the user experience questionnaire, all patients from the HF and IHD EG evaluated the system as an element of novelty, attractiveness, and perspicuity. Neutral and low scores were received for efficiency and stimulation, due to the technical problems they encountered while using it. The system usability scale score received was above 68 points in both study groups, which is considered the limit of acceptability, a result which highlights that patients appreciated the ease of use of the system. The TAM questionnaire results showed a mean score of 30.5 ± 4.08 for perceived usefulness, a mean score of 28.6 ± 3.35 for perceived ease of use, and a mean total score of 59.1 ± 7.43, in the HF EG; and a mean score of 29.8 ± 4.82 for perceived usefulness, a mean score of 29.6 ± 5.46 for perceived ease of use, and a mean total score of 59.1 ± 7.43, in the IHD EG.

## 4. Discussion

Cardiac rehabilitation is a central element of the secondary prevention of cardiovascular disease in patients, due to the benefits it has in terms of decreasing the morbidity-mortality rate, hospitalizations, and by improving effort tolerance and ensuring social reintegration. Despite its multiple benefits, there is a constant gap in the non-pharmacological treatment of cardiovascular patients, caused by the absence of continuity of care after discharge [28,29].

The pilot study we conducted on heart failure and ischemic heart disease Romanian patients aimed to evaluate the effectiveness and use of the vCare virtual assistant in their remote cardiac rehabilitation. In terms of exercise capacity, we can state that in the HF study group, the virtual assistant is at least equivalent to an ambulatory cardiac rehabilitation treatment. Considering the negative results obtained in the control group, we can consider it an extremely efficient alternative for patients who cannot access in-person cardiac recovery programs. In the IHD study group, although the results on exercise capacity were not as spectacular, they were also superior to not performing any cardiac recovery. The results highlighted in our study are in line with the study conducted by Chen and colleagues, where they observed that exercise tolerance, peak oxygen uptake, and quality of life may all be considerably improved by home-based cardiac rehabilitation in individuals with HF. Also, other data from other current specialized literature highlights the positive effect of telerehabilitation on the increased physical performance of cardiovascular patients. The patients’ perceived sense of monitoring increases their compliance with cardiac rehabilitation programs [30,31].

The virtual assistant also showed effectiveness in reducing cardiovascular risk factors, particularly high cholesterol levels and nicotine addiction, in both the HF and IHD study groups. Here, we consider the medical education service provided via the vCare app and the constant assistant–patient relationship to be of great help. Robotic medical education is a secondary prevention alternative that has been long discussed in the medical field as a solution to the gap between patients’ discharge and follow-up, especially for the elderly [32].

In terms of anxiety and depression levels, the experimental and ambulatory groups of patients experienced a reduction, whereas in the control groups there was no change. A recent review of telerehabilitation effects on heart failure patients found an improvement in patients’ depression and anxiety levels, physical capacity, and overall quality of life [33].

Further, the self-assessment of state of health (EQ-VAS) data collected at T_0_ and T_1_, suggested that patients in the vCare group had an important improvement in quality of life at the end of the rehabilitation program. Current literature suggests equivalence or a slight positive difference in quality of life between center-based cardiac rehabilitation and telerehabilitation, in favor of the latter [34,35,36].

In a systematic review and meta-analysis of telerehabilitation of heart failure patients, Cordeiro et al. highlighted that telerehabilitation enhances social skills, exercise tolerance, sexual activity, and heart failure symptoms, being at least as effective as center-based cardiac rehabilitation programs [37,38].

For ischemic heart disease patients, similar results from the current literature state that home-based cardiac recovery via remote-monitoring has as effective results as the classic cardiac rehabilitation programs. In their study, Escobar et al. used a remote electrocardiographic monitoring device for the remote-monitored group and stated no significant difference between the two methods in terms of exercise tolerance and recovery rate [39].

Based on our results, and those provided by the current literature, the virtual assistant seems a viable alternative for heart failure and ischemic heart disease patients to perform remote cardiac rehabilitation with the same benefits as a center-based one, from the comfort of their home [40].

## 5. Future Perspectives

Our results might serve as a stepping stone for future studies to come, that would explore the vCare system’s efficacy on larger groups of patients. Furthermore, our findings may assist policymakers in developing new legislation to incorporate this system in the non-pharmacological treatment of cardiac patients at home. In addition to the above stated points, the vCare system could further be improved by creating an application that is compatible with mobile phones, in order to facilitate its use.

## 6. Limitations

The study’s shortcomings are represented by the small sample size and the use of only the vCare tablet app. A more accurate understanding of the effectiveness of this method would have resulted from having an app that could operate cross-platform, on both tablets and phones and multiple operating systems.

## 7. Conclusions

In conclusion, cardiac recovery via a virtual assistant is possible, and the results on exercise capacity, cardiovascular risk factors, and quality of life are almost equivalent to those of classical rehabilitation.

## Figures and Tables

**Figure 1 ijerph-20-03937-f001:**
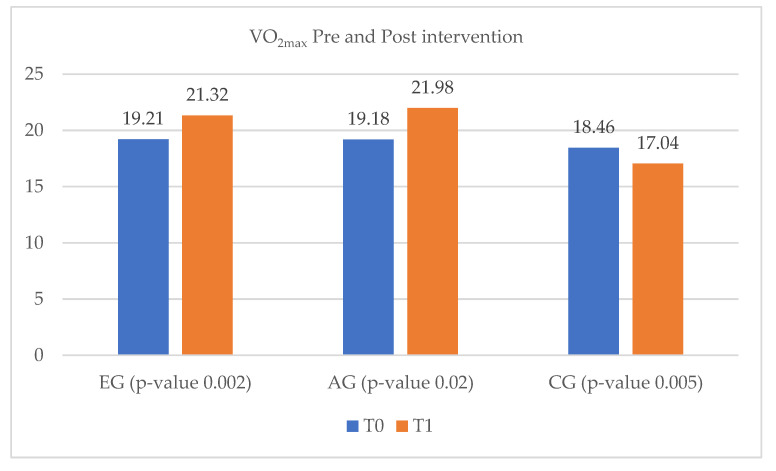
Values of VO_2max_ pre- and post-intervention. VO_2max_ = maximal oxygen consumption by the body (measured in mL/kg/min); EG = experimental group; AG = ambulatory group; CG = control group; T_0_ = before the rehabilitation period; T_1_ = after the rehabilitation period.

**Figure 2 ijerph-20-03937-f002:**
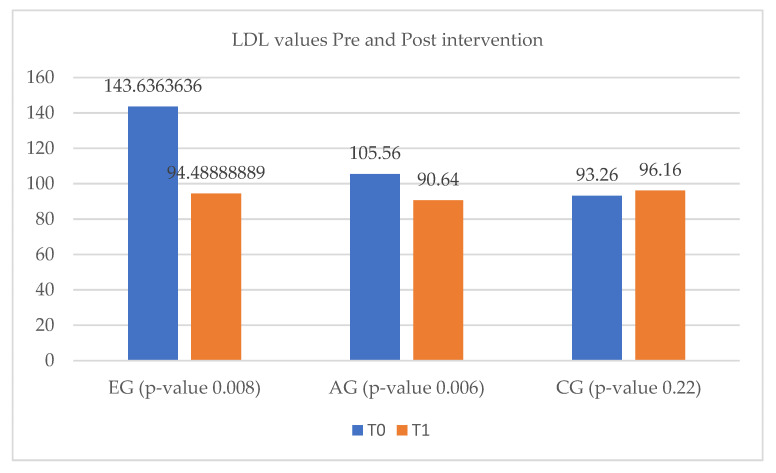
LDL cholesterol values in the heart failure study group. EG = experimental group; AG = ambulatory group; CG = control group; T_0_ = before the rehabilitation period; T_1_ = after the rehabilitation period.

**Figure 3 ijerph-20-03937-f003:**
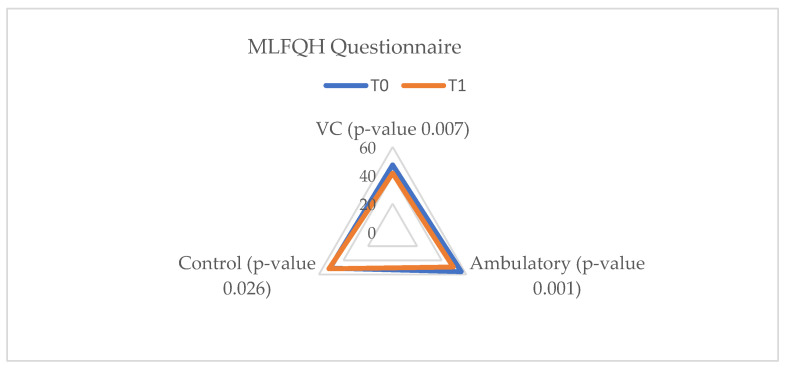
Minnesota questionnaire results. MLFQH = Minnesota living with heart failure questionnaire; EG = experimental group; AG = ambulatory group; CG = control group; T_0_ = before the rehabilitation period; T_1_ = after the rehabilitation period.

**Figure 4 ijerph-20-03937-f004:**
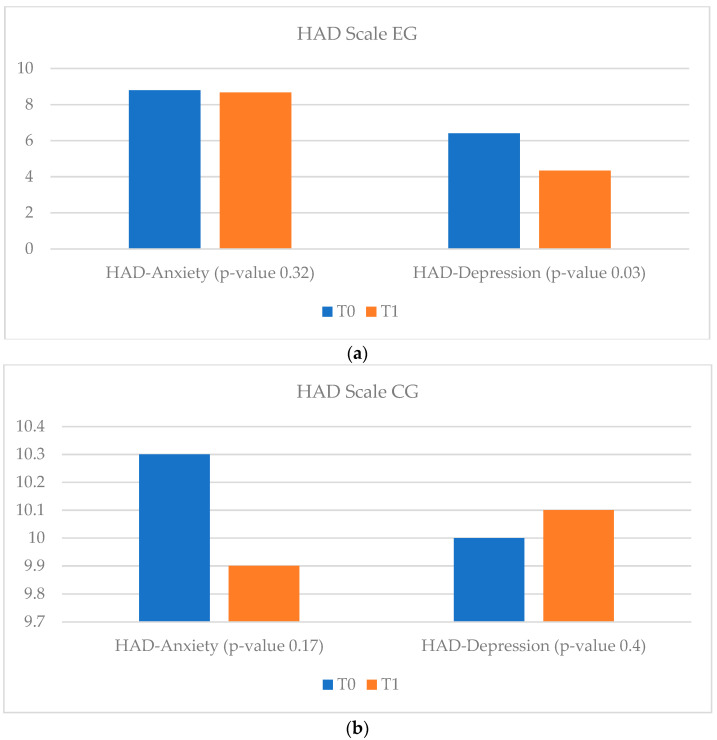
HAD scale results for HF EG and CG at T_0_ and T_1_. (**a**) HAD scale EG. (**b**) HAD scale CG. HAD = hospital anxiety and depression scale; HF= heart failure; EG = experimental group; CG = control group; T_0_ = before the rehabilitation period; T_1_ = after the rehabilitation period.

**Figure 5 ijerph-20-03937-f005:**
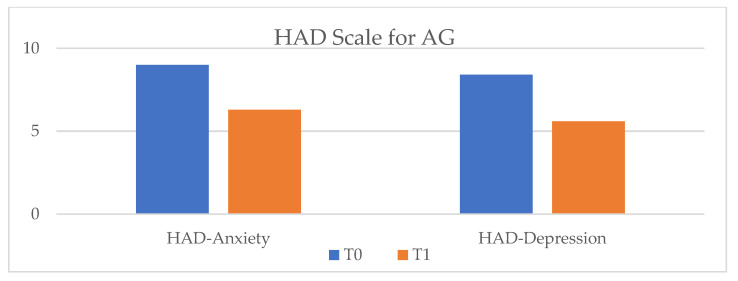
HAD scale results for AG at T_0_ and T_1_. HAD = hospital anxiety and depression scale; AG = control group; T_0_ = before the rehabilitation period; T_1_ = after the rehabilitation period.

**Figure 6 ijerph-20-03937-f006:**
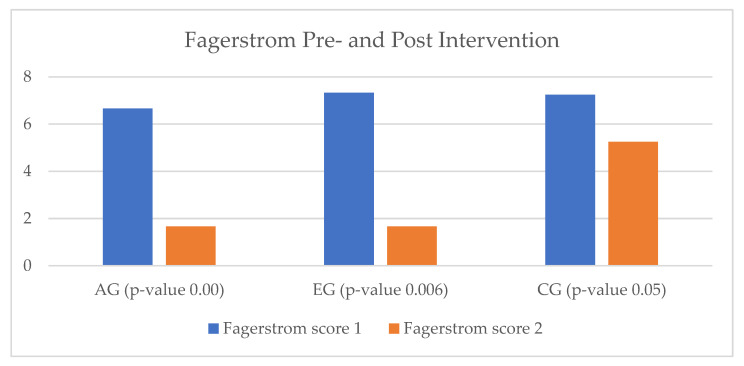
Smoking status of heart failure patients. HF = heart failure; AG = ambulatory group; EG = experimental group; CG = control group; T_0_ = before the rehabilitation period; T_1_ = after the rehabilitation period.

**Figure 7 ijerph-20-03937-f007:**
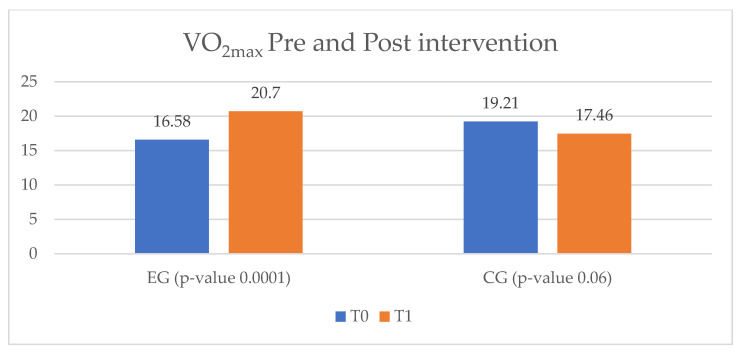
VO_2max_ pre- and post-intervention. VO_2max_ = maximal oxygen consumption by the body (measured in mL/kg/min); EG = experimental group; CG = control group; T_0_ = before the rehabilitation period; T_1_ = after the rehabilitation period.

**Figure 8 ijerph-20-03937-f008:**
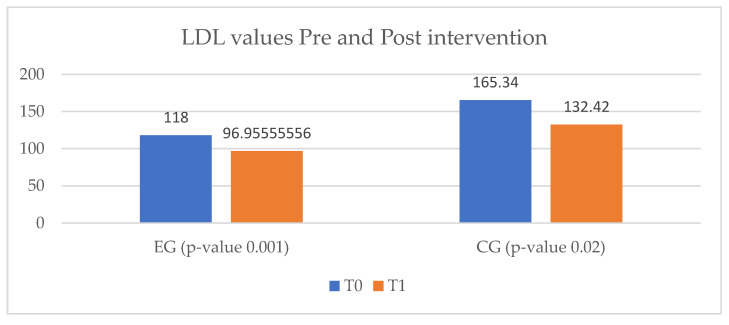
LDL-cholesterol values in the ischemic heart disease study group. EG = experimental group; CG = control group; T_0_ = before the rehabilitation period; T_1_ = after the rehabilitation period.

**Figure 9 ijerph-20-03937-f009:**
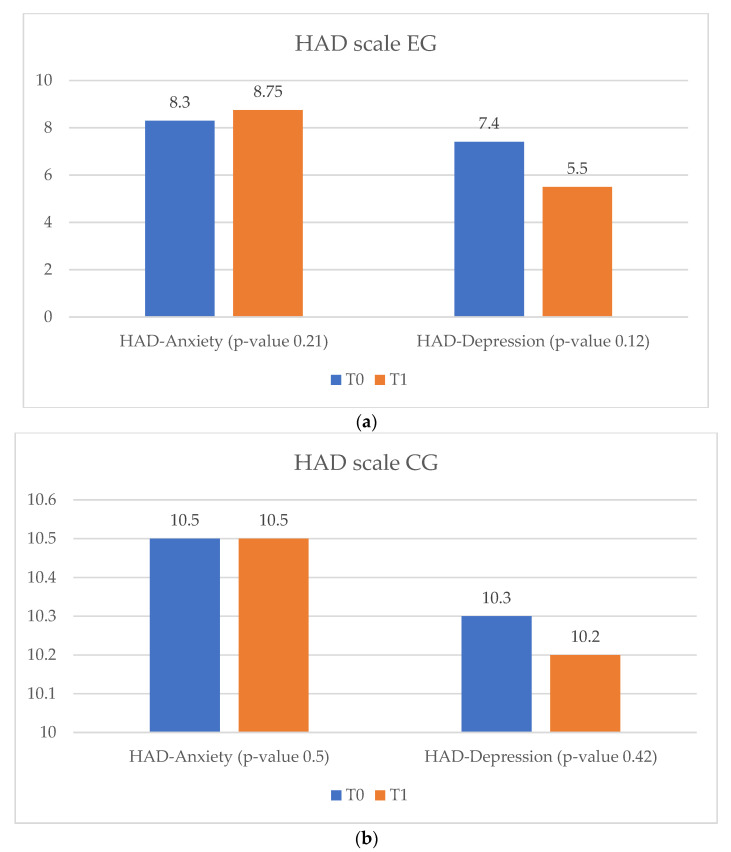
HAD scale results for IHD EG and CG at T_0_ and T_1_. (**a**) HAD scale EG. (**b**) HAD scale CG. HAD = hospital anxiety and depression scale; IHD = ischemic heart disease; EG = experimental group; CG = control group; T_0_ = before the rehabilitation period; T_1_ = after the rehabilitation period.

**Figure 10 ijerph-20-03937-f010:**
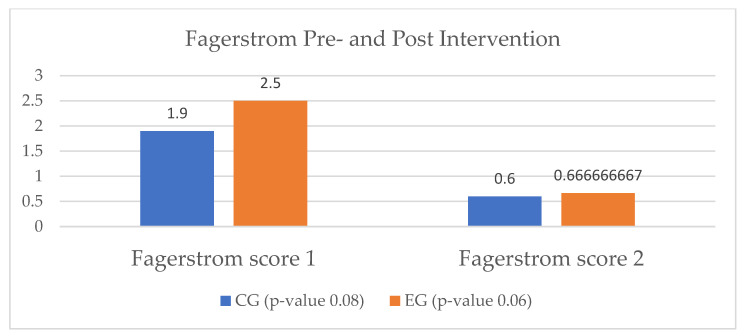
Smoking status of IHD patients. IHD = ischemic heart disease; EG = experimental group; CG = control group; T_0_ = before the rehabilitation period; T_1_ = after the rehabilitation period.

**Figure 11 ijerph-20-03937-f011:**
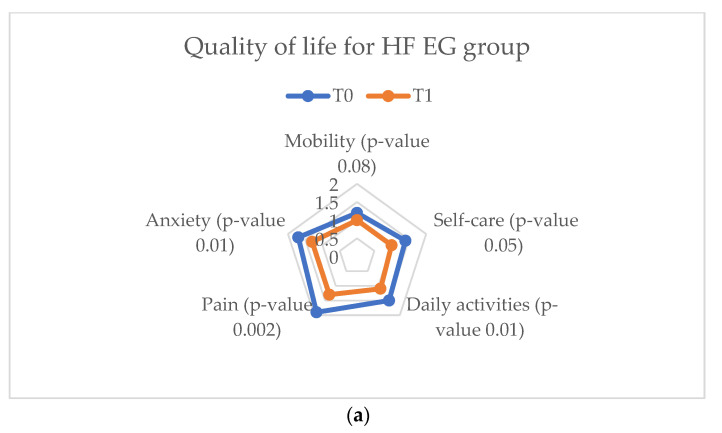
EuroQol-5D results parameters for the heart failure study groups. (**a**) Quality of life for HF EG group. (**b**) Quality of life for HF AG group. (**c**) Quality of life for HF CG group.

**Figure 12 ijerph-20-03937-f012:**
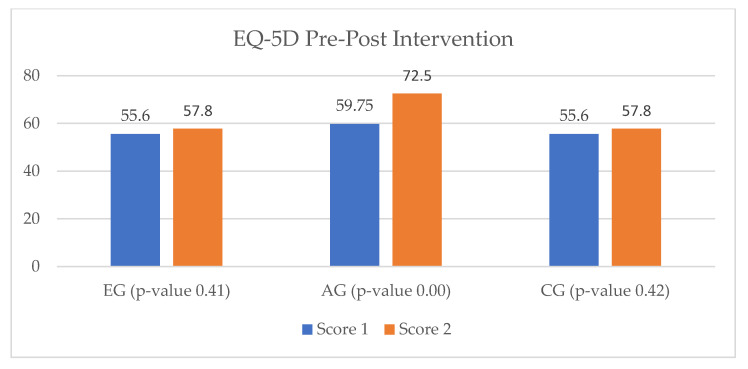
EQ-VAS results for the heart failure study group.

**Figure 13 ijerph-20-03937-f013:**
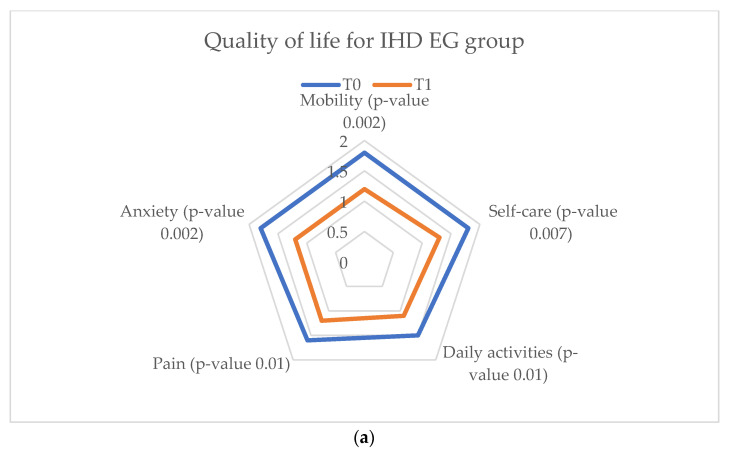
EuroQol-5D results parameters for the ischemic heart disease study group. (**a**) Quality of life for IHD EG. (**b**) Quality of life for IHD CG.

**Figure 14 ijerph-20-03937-f014:**
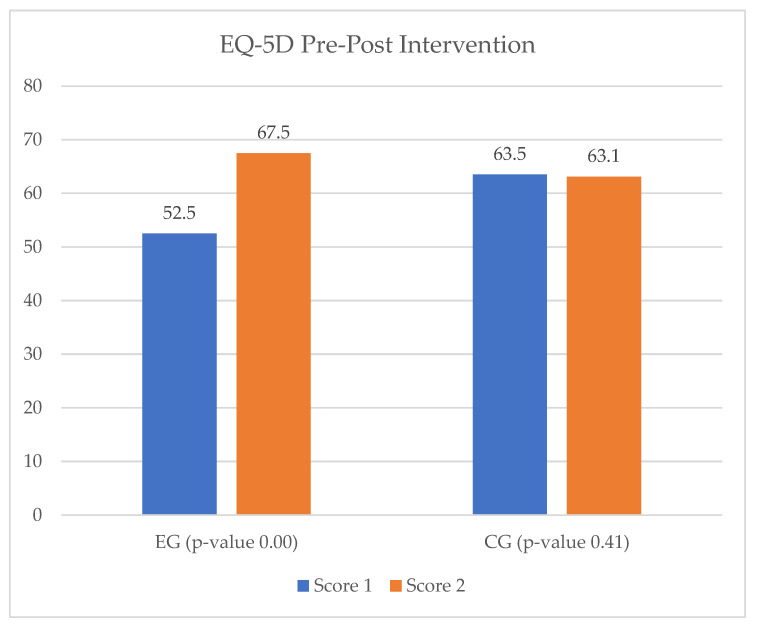
EQ-VAS results for the ischemic heart disease study group.

**Table 1 ijerph-20-03937-t001:** Inclusion and exclusion criteria.

Inclusion Criteria	Exclusion Criteria
Age > 50 years old	Unstable condition requiring treatment
Heart failure class NYHA II-III	Unstable angina
History of myocardial infarction	Uncontrolled sustained arrhythmias
History of unstable/stable angina pectoris	Severe uncontrolled arterial hypertension
History of percutaneous coronary intervention	Active myo-/peri-/endocarditis
History of coronary artery by-pass grafting	Recent pulmonary embolism
History of heart failure and/or ischemic heart disease therapy	Inability to understand and comply with protocol and/or give informed consent

**Table 2 ijerph-20-03937-t002:** Questionnaires used for the psychological evaluation of enrolled patients.

Minnesota Living with Heart Failure	HADS Scale (Anxiety and in-Hospital Depression)	Fagerstrom Test for Nicotine Dependence	EuroQol-5D (EQ5D)	Health Self-Evaluation Scale (EQ-VAS)
a self-administered, 21-item disease-specific instrument which assesses the quality of life among patients with heart failure; used only for HF use case [20]	an assessment tool consisting of a 14-item measure designed to assess anxiety and depression symptoms in medical patients, with an emphasis on reducing the impact of physical illness on the total score [21]	a standard assessment tool for the intensity of physical addiction to nicotine; its purpose is to provide an ordinal measure of nicotine dependence related to cigarette smoking [22]	a standard assessment quality of life questionnaire which has five dimensions and a score range from zero (no problems) to four (inability to walk, inability to perform daily activities, extreme pain, or extreme anxiety/depression), or related to their perceived health from zero to one hundred percent, with zero representing the worst possible health and one hundred percent the best possible [23]	a standard self-evaluation scale of a patient’s health, with two endpoints of‚ “The best health you can imagine’’ and‚ “The worst health you can imagine’’; it is used to reflect the patient’s judgement [24]

**Table 3 ijerph-20-03937-t003:** List of the equipment used in HF and IHD use cases.

Device	Model
Wristband	XIAOMI Mi Band 4
Weight scale	XIAOMI weight scale
Blood pressure device	Beurer BM85
Camera	Astra Orbbec camera
Software license-camera	Nuitrack software version v.0.36.7
Tablet	Lenovo Tab M10 Full HD Plus
Television (TV)	Owned by the subject
Set top box	Android TV BOX X99 4k Ultra HD

**Table 4 ijerph-20-03937-t004:** Monitoring parameters of the devices used.

Device	Parameters
Wristband	Heart rate and number of steps
Weight scale	Weight
Blood pressure device	Blood pressure
Camera	Motion recognition
Tablet	Interaction with the patient
Television (TV)	Running of the serious games
Set top box	Ultra-modern personal computer with component integration role

**Table 5 ijerph-20-03937-t005:** Questionnaires used for the usability perception of the vCare experience.

User Experience Questionnaire (UEQ)	System Usability Scale (SUS)	Technology Acceptance Model (TAM)
was administered to measure classical usability aspects and user experience aspects. It is a questionnaire composed of 26 items built as pairs of contrasting attributes [25]	is a questionnaire that consists of 10 items, with five response options for each item (from “strongly disagree” to “strongly agree”), which allows the subjective evaluation of the usability of the system under examination after the direct interaction of the user with the system [26]	is the most popular model among those proposed to explain and predict the acceptance of a system [27]

**Table 6 ijerph-20-03937-t006:** Components of the virtual cardiac rehabilitation program.

E-learning	Aerobic Physical Activity	Resistance Training	Medication Intake Support	Vital Stats Control	Smoking Cessation Activity	Anxiety and Depression Reduction	Alcohol Reduction
Medical education	Daily number of steps	Strengh training with serious games	Interaction confirming the administration of pharmacological therapy	Vital parameters monitoring	Assessment of the number of cigarettes consumed	Assessment of anxiety and depression status	Assessment of the amount of alcohol consumed

**Table 7 ijerph-20-03937-t007:** vCare steps parameter and active weeks for the HF subjects.

	Active Days	Adherence	Average Steps per Active Days	Active Weeks	Adherence	Average Accesses per Active Weeks
Patient1	3	8%	211	4	72%	3
Patient2	7	30%	1467	3	91%	2
Patient4	22	25%	8632	10	79%	6
Patient5	1	2%	704	2	32%	5
Patient6	7	6%	5251	13	80%	5
Patient7	3	4%	524	3	31%	2
Patient8	3	10%	474	4	93%	3
Patient9	35	38%	2545	14	93%	10
Patient10	20	23%	7863	13	80%	5
mean	10.1	16%	2476	6.6	71%	4
std	12	0.1	2993	4.9	0.2	3

**Table 8 ijerph-20-03937-t008:** vCare steps parameter and active weeks for the IHD subjects.

	Active Days	Adherence	Average Steps per Active Days	Active Weeks	Adherence	Average Accesses per Active Weeks
Patient1	1	4%	514	3	84%	5
Patient2	3	13%	3574	3	84%	5
Patient4	3	6%	1093	4	31%	2
Patient5	83	91%	4208	15	100%	13
Patient6	31	34%	4049	11	85%	2
Patient7	9	22%	1451	6	100%	5
Patient8	30	33%	4860	5	73%	9
Patient9	12	27%	3000	3	82%	7
Patient10	21	30%	3930	4	40%	4
mean	19.3	26%	2668	6.7	67.9%	5.2
std	29.9	0.3	1579	4.6	0.2	4

## Data Availability

Data available upon request due to ethical restrictions. The data presented in this study are available upon request from the corresponding author. The data are not publicly available due to ethical restrictions.

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
