# Peer review of "Assessing the Efficacy of a Virtual Assistant in the Remote Cardiac Rehabilitation of Heart Failure and Ischemic Heart Disease Patients: Case-Control Study of Romanian Adult Patients"

_ijerph, 2023, doi:10.3390/ijerph20053937_

Round 1
Reviewer 1 Report
Dear Editor,
Thank you for the opportunity to review the article "Where should we go forward: conventional or telerehabilitation? With the aim of building and implementing the virtual assistant in patients' homes and evaluating the effectiveness of the tele-rehabilitation programme.
The following changes are suggested:
Title:
- Include more information about the target population and type of study.
Methodology:
Add information about the vCARE project,
Explain the programme in detail
Explain the measures of randomisation and blinding of participants and assessors.
Add the ethical issues in the methodology and procedures
Results:
-Add the P value to each of the graphs and their results,
Discussion:
Explore the different results in the light of other studies.
Author Response
Dear Reviewer,
First I would like to thank you for the time offered to review our manuscript. Thank you for your suggestions, as they help us improve the quality of our work.
I will list point by point the changes made to the initially submitted manuscript.
- The title has been changed into a more adequate one;
- We have described in detail the vCare project and its programme;
- We have described the measures of randomisation and blinding of participants and assessors;
- We have added the ethical issues in the methodology and procedures;
- We have calculated the p-value for every parameter analysed;
- We have enhanced the discussion section with 2 more studies regarding the remote-cardiac rehabilitation programs;
Sincerely,
M.P
Reviewer 2 Report
1. The title of the manuscript is quite unspecific and does not give a proper idea about the manuscripts content and the study meaning and purpose. I suggest the authors to change the title and rewrite it to provide a clear view of the manuscript content.
2. It would be interesting if the authors add a brief discussion comparing with other studies or results from the current literature in the Discussion.
Author Response
Dear Reviewer,
First I would like to thank you for the time offered to review our manuscript. Your suggestions are valuable as they help us improve our work.
I will list point by point the changes we have performed on the initial submitted version of the manuscript.
- The title has been changed to a more adequate one;
- We have detailed the vCare project and its programme;
- We have detailed the methodology of our study;
- We have enhanced the Discussion chapter with more studies regarding the remote-cardiac rehabilitation programs;
Sincerely,
M.P
Reviewer 3 Report
In this study, the clinical use of the vCare system was evaluated by enrolling 50 patients. The subject of this manuscript is clinical trials, and it is recommended to revise the title and manuscript framework in strict accordance with the clinical research paradigm.
There are some major issues with this paper, and the research authors are requested to make improvements,
1. Supplement the specific content of vCare and enumerate what services the system can provide to patients. The content in Table 3 cannot explain the problem.
2. Describe the research methodology and statistical methods in detail.
3. All important statistical results should be standardized reporting values.
Author Response
Dear Reviewer,
First I would like to thank you for the time offered to review our manuscript. Your suggestions are of great value as they help us improve our work.
I will list point by point the changes we have performed on the initial submitted manuscript:
- We have revised the title of the manuscript and the manuscript framework;
- We have detailed the vCare project and its programme in the Methodology section, as well as the services the system offers;
- We have described the research methodology and results in more detail;
- We have adjusted the statistical results;
- We have enhanced the Discussion chapter with more results from the current literature regarding the remote-cardiac rehabilitation programs.
Sincerely,
M.P
Round 2
Reviewer 1 Report
Dear Editor,
Congratulations to the authors for their efforts in improving the article.
I would also suggest the following changes:
Title include method
Review table formatting
I see no relevance in charts 1 and 2. I suggest removing them.
Author Response
Dear Reviewer,
Thank you for your suggestions.
I have made the appropriate changes suggested as follows:
- I have changed the title into a more adequate one which containts the method of the study and the targeted population;
- I have removed charts 1 and 2 for both study groups: IHD and HF.
- I have reviewed the table formatting.
Sincerely,
M.P
Reviewer 3 Report
The author has made a large revision, but this manuscript is still not standard as a clinical research report.
The authors should consider the STROBE statement (Strengthening the Reporting of Observational studies in Epidemiology) and make sure that this manuscript follows the standards for reporting observational studies outlined therein.
Author Response
Dear Reviewer,
Thank you for your suggestions. We have used the STROBE statement and accordingly we have performed the following changes to the manuscript:
- changed the title into a more adequate one;
- we have rearranged and enhanced the information from the Materials and Methods chapter, which is now structured into:
- Study design and objectives;
- Profile of enrolled patients;
- Ethical considerations;
- Initial assessment and structuring of the virtual environment;
- 3-month assessment and collected parameters;
- Virtual assistant components;
- Statistical analysis;
- Furthermore, in the Results Chapter, have been defined social, demographic, and extensive medical parameters of the enrolled patients; have been offered details about the recruitment period, excluded patients and dropped-out patients;
- In the Discussion chapter we have described the objectives obtained and compared them to more suitable results from the literature.
- The Conclusion chapter has been transformed into a more specific one, containing the main conclusion of our study.
Sincerely,
M.P